# Prebiotics Improve Blood Pressure Control by Modulating Gut Microbiome Composition and Function: A Systematic Review and Meta-Analysis

**DOI:** 10.3390/nu17152502

**Published:** 2025-07-30

**Authors:** Abdulwhab Shremo Msdi, Elisabeth M. Wang, Kevin W. Garey

**Affiliations:** Department of Pharmacy Practice and Translational Research, College of Pharmacy, University of Houston, 4349 Martin Luther King Boulevard, Houston, TX 77204, USA; emwang@central.uh.edu (E.M.W.); kgarey@central.uh.edu (K.W.G.)

**Keywords:** prebiotic, symbiotic, microbiome, short-chain fatty acids, human health, hypertension

## Abstract

**Background:** Ingestion of dietary fibers (DFs) is a safe and accessible intervention associated with reductions in blood pressure (BP) and cardiovascular mortality. However, the mechanisms underlying the antihypertensive effects of DFs remain poorly defined. This systematic review and meta-analysis evaluates how DFs influence BP regulation by modulating gut microbial composition and enhancing short-chain fatty acid (SCFA) production. **Methods:** MEDLINE and EMBASE were systematically searched for interventional studies published between January 2014 and December 2024. Eligible studies assessed the effects of DFs or other prebiotics on systolic BP (SBP) and diastolic BP (DBP) in addition to changes in gut microbial or SCFA composition. **Results:** Of the 3010 records screened, nineteen studies met the inclusion criteria (seven human, twelve animal). A random-effects meta-analysis was conducted on six human trials reporting post-intervention BP values. Prebiotics were the primary intervention. In hypertensive cohorts, prebiotics significantly reduced SBP (−8.5 mmHg; 95% CI: −13.9, −3.1) and DBP (−5.2 mmHg; 95% CI: −8.5, −2.0). A pooled analysis of hypertensive and non-hypertensive patients showed non-significant reductions in SBP (−4.5 mmHg; 95% CI: −9.3, 0.3) and DBP (−2.5 mmHg; 95% CI: −5.4, 0.4). Animal studies consistently showed BP-lowering effects across diverse etiologies. Prebiotic interventions restored bacterial genera known to metabolize DFs to SCFAs (e.g., *Bifidobacteria*, *Akkermansia*, and *Coprococcus*) and increased SCFA levels. Mechanistically, SCFAs act along gut–organ axes to modulate immune, vascular, and neurohormonal pathways involved in BP regulation. **Conclusions:** Prebiotic supplementation is a promising strategy to reestablish BP homeostasis in hypertensive patients. Benefits are likely mediated through modulation of the gut microbiota and enhanced SCFA production.

## 1. Background

Hypertension (HTN) is a leading global cause of cardiovascular morbidity and mortality, affecting 30–50% of adults worldwide [1,2,3]. Despite the availability of effective pharmacotherapies, only one in five adults achieve adequate blood pressure (BP) control [1]. Therefore, prevention strategies are prioritized by the World Health Organization and endorsed by clinical guidelines [4,5]. Specifically, dietary fiber (DF) is recommended as a safe and effective intervention for BP regulation. Meta-analyses of randomized controlled trials (RCTs) showed that DF supplementation can lower BP by approximately 5 mmHg, especially in individuals with HTN [6]. Two recent meta-analyses demonstrated a linear, dose-dependent reduction in mortality with higher DF intake, independent of pharmacologic therapy [7,8]. These benefits were more pronounced in hypertensive patients without cardiovascular disease (CVD), underscoring the potential role of DF in preventing CVD-related morbidities. Yet, the mechanisms by which DF improves BP remain poorly understood.

The gut microbiome is a key modulator of BP regulation. Pre-hypertensive and hypertensive individuals exhibit distinct microbial profiles characterized by reduced abundance of beneficial anaerobic, SCFA-producing genera, including *Roseburia*, *Faecalibacterium*, *Bifidobacterium*, *Coprococcus*, and *Akkermansia* [9,10,11]. Importantly, fecal microbiota transplantation (FMT) from hypertensive patients induces both dysbiosis and elevated BP in germ-free mice, suggesting a causal role of the microbiota in HTN [9]. Highly fermentable DFs like inulin and resistant starch (RS), modulate the gut microbial composition and promote SCFA production in the human colon [12,13]. SCFAs (acetate, propionate, and butyrate) are microbial metabolites that exert local and systemic effects by interacting with G-protein coupled receptors (GPCRs) across multiple organs [14]. Butyrate improves gut barrier integrity and reduces systemic and neuroinflammation, while acetate and propionate modulate peripheral resistance, renin-angiotensin signaling, and tissue remodeling in the heart and kidneys [14,15,16,17]. All these effects are associated with improved BP homeostasis.

Despite evidence that DF lowers BP and improves cardiovascular outcomes, the underlying mechanisms remain poorly understood, particularly for microbiome-related pathways [18]. This limited mechanistic insight likely contributes to the absence of strong, guideline-based recommendations for DF supplementation in HTN management. To address this gap, we conducted a systematic review to evaluate how DF affects BP by modulating gut microbial composition and function (SCFA production), synthesizing evidence from both human and animal studies to clarify microbiome-related mechanisms.

## 2. Methods

This systematic review and meta-analysis was conducted in accordance with the Preferred Reporting Items for Systematic Reviews and Meta-Analyses (PRISMA) guidelines [19]. Figure 1 illustrates the study selection process. Studies were included if they met all of the following eligibility criteria: (1) interventional design examining the effects of prebiotic or DF interventions on BP; (2) reported outcomes related to gut microbial composition or function, including measures of alpha or beta diversity, specific bacterial taxa, or SCFA levels (acetate, butyrate, or propionate); (3) publication within the last 10 years (from 1 January 2014 to 16 December 2024); and (4) published in English. We excluded abstracts, systematic reviews, meta-analyses, post hoc analyses, and methodological reports. In addition to primary outcomes, secondary data were extracted regarding the type, dose, and duration of the DF intervention, as well as its effects on gut, immune, cardiovascular, renal, or central nervous system (CNS) health, to provide mechanistic insights.

MEDLINE (PubMed) and EMBASE (Elsevier) were systematically searched from 1 January 2014 to 16 December 2024. Key terms included the following: Dietary Fiber, Prebiotic, Microbiome Composition, Bacterial Composition, Gastrointestinal Microbiome, Bacterial Taxa, Short-Chain Fatty Acids, Volatile Fatty Acids, Acetate, Butyrate, and Propionate. The complete search strategy is detailed in Appendix A. All retrieved citations were imported into Rayyan, a web-based platform for systematic review screening [20]. Two reviewers (ASM and EMW) independently screened titles and abstracts using predefined eligibility criteria. Full-text articles of potentially eligible studies were then assessed for inclusion based on the same criteria. Data extraction was performed independently using a standardized template, and discrepancies were resolved through discussion among reviewers (ASM, EMW, and KWG).

We performed a meta-analysis to quantify the pooled effect of prebiotic supplementation on systolic blood pressure (SBP) and diastolic blood pressure (DBP) in clinical trials. A random-effects model was used to compute the pooled effect size while accounting for between-study heterogeneity. The effect size was calculated as mean differences (MD) in post-intervention BP values between intervention and control groups using the generic inverse variance method, as recommended in the Cochrane Handbook for Systematic Reviews of Interventions [21]. Heterogeneity was quantified using the Q statistic and the *I*^2^ index [21]. To explore sources of heterogeneity, we conducted a mixed-effects meta-regression to assess whether moderators, including population type (e.g., hypertensive, metabolic syndrome), DF type, dose, and duration, explained the variability among studies. Publication bias was assessed using Egger’s regression test. All analyses were performed in R (version 4.3.2; R Foundation for Statistical Computing, Vienna, Austria) using the *metafor* package.

## 3. Results

After removing duplicates, we screened the titles and abstracts of 3010 records and identified 41 studies for full-text review. Of these, 13 studies were excluded for not reporting changes in BP, gut microbial composition, or microbial function. An additional eight studies were excluded because they were available only as abstracts, and one study was excluded for using a non-fiber intervention. Nineteen studies met all the inclusion criteria and were included in the final synthesis. Table 1 and Table 2 summarize the characteristics and main findings of the human and animal studies included, respectively. Risk of bias assessments are shown in Appendix A.

### 3.1. Clinical Data

#### 3.1.1. Overview of Included Studies

Seven human studies were included in the analysis (Table 1). All were pre-planned RCTs except for one exploratory secondary analysis of an RCT [22,23,24,25,26,27,28]. The investigated population primarily consisted of overweight adults with metabolic syndrome (mean BP ~130/82 mmHg) or HTN (mean BP ~140/90 mmHg). The main intervention was prebiotic supplementation, inulin, or RS, with or without dietary counseling. Prebiotic doses ranged from 9 to 40 g/day over 2 to 12 weeks.

#### 3.1.2. Effect on Blood Pressure

Prebiotic supplementation was associated with consistent BP reduction in patients with preexisting HTN. Of the seven clinical trials, six were included in the meta-analysis; one was excluded due to missing post-intervention BP values [27]. A pooled analysis showed a reduction in SBP by −4.5 mmHg (95% CI: −9.3, 0.3; *p* = 0.07) and in DBP by −2.5 mmHg (95% CI: −5.4, 0.4; *p* = 0.09); however, these reductions did not reach statistical significance (Figure 2A,B). Heterogeneity was substantial (*I*^2^ ≈ 80% for SBP). Meta-regression identified population type as a key source of variability. In a subgroup analysis restricted to hypertensive cohorts (*n* = 3 studies), prebiotics significantly reduced SBP (−8.5 mmHg, 95% CI: −13.9, −3.1; *p* = 0.002) and DBP (−5.2 mmHg, 95% CI: −8.5, −2.0; *p* = 0.002), with moderate heterogeneity (*I*^2^ ≈ 46%) (Figure 2C,D). No evidence of publication bias was detected (Egger’s test, *p* = 0.07). Most participants were already receiving antihypertensive therapy. In one trial, oat bran combined with dietary counseling reduced peak SBP and lowered antihypertensive medication use [23]. In untreated hypertensive patients, 40 g/day of esterified RS reduced 24 h SBP by 6 mmHg [24]. In contrast, two studies conducted in patients with metabolic syndrome reported no treatment effect [25,26]. In both studies, BP was a secondary outcome, and the control groups received guideline-based diets, which may have limited the power to detect an effect, if one existed.

#### 3.1.3. Mechanistic Insights

To explore the underlying mechanisms, we examined changes in gut microbiota and SCFA levels. Prebiotic supplementation significantly altered gut microbial composition, as assessed by Bray–Curtis beta diversity [23,24]. Across six studies, prebiotics increased the abundance of strict anaerobic butyrate producers from the Firmicutes phylum, particularly the Clostridia class. Notably, interventions enriched the Lachnospiraceae and Ruminococcaceae families, including genera such as *Anaerostipes*, *Ruminococcus*, and *Coprococcus*. Inulin-based fibers also increased *Bifidobacterium* and *Akkermansia*, genera associated with anti-inflammatory activity [27,28]. Four studies assessed SCFA profiles by sampling site-dependent variations. Plasma butyrate levels increased consistently across three studies [22,24,28]. In contrast, one study reported a reduction in fecal propionate percentage following prebiotic supplementation, which may reflect improved epithelial integrity and higher systemic absorption [25].

Several studies reported improvements in other classical cardiovascular risk markers, including peripheral resistance, body weight, and lipid and glucose profiles [22,25,26]. In healthy volunteers consuming low-fiber diets, inulin supplementation lowered GlycA levels, a novel inflammatory biomarker of cardiovascular risk [28]. Notably, plasma SCFA levels explained up to 75% of the variation in cardiovascular markers, underscoring their potential mechanistic role [28]. Finally, prebiotic supplementation improved gut barrier integrity, reduced endotoxemia, and dampened inflammatory responses compared to a placebo [25,27,28].

### 3.2. Animal Data

#### 3.2.1. Overview of Included Studies

We included 12 animal studies investigating various hypertensive phenotypes: angiotensin-II (Ang-II)-induced HTN (*n* = 3), programmed developmental HTN (*n* = 2), hypoxia-induced HTN (*n* = 2), metabolic HTN (*n* = 2), salt-sensitive HTN (*n* = 2), and spontaneous HTN (*n* = 1) (Table 2) [29,30,31,32,33,34,35,36,37,38,39,40]. Most studies used RS or inulin-type fructans (ITF) as the prebiotic intervention. One study employed a symbiotic combination of inulin with *Lactobacillus*, *Bifidobacterium*, and *Streptococcus* strains [36].

#### 3.2.2. Effect on Blood Pressure

Prebiotic supplementation reduced BP across all models, regardless of the underlying etiology. The only exception was O’Connor et al., in which ITF failed to improve BP in a mild intermittent hypoxia model [35]. This model did not alter the gut microbial composition, suggesting a microbiome-independent mechanism.

#### 3.2.3. Mechanistic Insights

Prebiotics consistently shifted the gut microbiota toward an improved health-associated profile across all hypertensive phenotypes. Most studies (10/12) reported an increased abundance of strict anaerobes and SCFA-producing taxa, such as genus *Bacteroides*, genus *Bifidobacterium*, and genera within the Lachnospiraceae family. In parallel, prebiotic supplementation reduced Gram-negative facultative taxa, including the order Enterobacterales and the genera *Prevotella*, and *Alistipes*. Four of five studies assessing the Firmicutes/Bacteroidetes (F/B) ratio observed a reduction following prebiotic intervention, a microbial signature associated with lower BP [30,33,37,38,40].

Taxonomic changes varied by hypertensive phenotype, but SCFA levels increased consistently, regardless of the hypertensive model. In renin–angiotensin–aldosterone system (RAAS)-driven HTN (Ang-II infusion or salt-loading models), RS stabilized microbial composition by increasing the genera *Bacteroides*, *Bifidobacterium*, and the family Lachnospiraceae, while suppressing Enterobacterales and *Alistipes*. These changes preserved acetate and propionate levels and attenuated Ang-II-induced dysbiosis, inflammation, and cardiac remodeling [29,30,31,39]. In developmental HTN, ITF supplementation, administered to either the mother or offspring, restored *Akkermansia muciniphila* and SCFA production [32,33]. In hypoxia-induced HTN (a model of obstructive sleep apnea), RS increased the genera *Bifidobacterium*, *Blautia*, and acetate levels. Acetate colonic infusion directly reduced BP and inflammation, underscoring the importance of local SCFA production in this phenotype [34]. Notably, the effectiveness of each prebiotic depended on hypoxia severity. Mild hypoxia did not trigger microbial shifts, supporting a limited microbiome role in this etiology [35]. In metabolic HTN, prebiotic or symbiotic supplementation increased phylum Actinobacteria and family Lachnospiraceae, while reducing the phylum Proteobacteria. These microbial shifts elevated both local and systemic SCFA exposure [36,37].

Several studies established a causal link between the microbiome and BP regulation. FMT from hypertensive, low-fiber-fed mice induced HTN and cardiac injury in germ-free recipients [30]. Supplementation with acetate or butyrate alone reproduced the cardiovascular benefits seen with prebiotics [30,32,34,37,38]. Conversely, antibiotics abolished the antihypertensive effects of prebiotic supplementation, further supporting the microbiome’s role [37]. Mechanistically, prebiotics improved gut barrier function in 10 out of 12 studies, enhancing tight junction expression and reducing endotoxemia and inflammation. In over half the studies (7/12), prebiotics modulated the gut–heart and/or gut–kidney axes, attenuating tissue fibrosis and hypertrophy. In developmental and hypoxia-induced HTN, prebiotics modulated the gut–CNS axis, reducing neuroinflammation.

## 4. Discussion

DF intake is a safe, accessible intervention associated with reduced BP and cardiovascular mortality; however, the mechanisms for these beneficial effects have not been well elucidated. This review synthesized clinical and preclinical evidence to evaluate how DF regulates BP through gut microbial modulation. Our findings suggest that highly fermentable DFs (prebiotics) promote SCFA production, reduce inflammation, and modulate gut–organ axes to maintain BP homeostasis. Across controlled trials, prebiotic supplementation demonstrated greater and more consistent BP reductions in hypertensive individuals compared to those with normal BP [6,8]. This finding aligns with our results, showing that prebiotic supplementation led to an 80% greater reduction in SBP for hypertensive patients (−8.5 mmHg) compared to other cohorts (−4.5 mmHg). Moreover, there was a notable reduction in the effect heterogeneity, with the *I*^2^ index decreasing from 80% to 45%. Notably, selective reductions in peak SBP values without lowering minimum SBP were observed in patients with stage I HTN, suggesting a homeostatic rather than hypotensive effect [23]. Animal studies confirmed consistent BP-lowering effects across diverse hypertensive models, supporting a mechanistic role for prebiotics independent of the underlying etiology. This is important, as elevated BP is often maintained by multiple pathological pathways. These effects are likely attributable to the ability of prebiotics to restore gut microbial structure and function, which is crucial for regulating key pathological pathways related to HTN, and resting BP.

Prebiotic supplementation restored the baseline abundance of SCFA-producing genera, namely, *Bifidobacteria*, *Lactobacillus*, *Akkermansia*, *Coprococcus*, and *Anaerostipes*, all previously associated with reduced BP [11]. *Bifidobacteria* and *Lactobacillus* are known for their roles in producing acetate and propionate, and they demonstrate anti-inflammatory and antihypertensive effects, particularly in individuals with salt-sensitive HTN [39,41,42]. *Akkermansia* contributes to propionate production, mucin recycling, and epithelial barrier integrity [43,44]. *Coprococcus* and *Anaerostipes*, both Firmicutes that belong to the Lachnospiraceae family, are major butyrate producers with roles in gut–immune regulation [45]. In contrast, prebiotic supplementation diminished key taxa associated with proinflammation and hypertension, such as the genera *Prevotella*, *Alistipes*, and the order Enterobacterales [11]. Importantly, across human and animal studies, SCFA levels increased despite variation in taxonomic shifts. Additionally, direct sodium butyrate supplementation (600 mg/day) reduced TNF-α and BP in diabetic patients [27]. This suggests that SCFA production may serve as a more consistent indicator of gut functional response than microbial composition alone. It is highly likely that SCFAs act locally and distally to maintain cardiovascular hemostasis.

SCFAs interact with GPCRs along gut–organ axes to reduce inflammation, vascular resistance, and tissue remodeling (Figure 3). Acetate and propionate lower colonic pH, activating proton-sensitive GPR65, which lowers CD8+ T cell activity and fibrosis in cardiac and renal tissues [31]. Interestingly, a high colonic pH is correlated with an elevated BP in clinical studies [24,31]. Acetate also downregulates *Egr1*, a master transcriptional regulator of cardiac and renal inflammation, and signals via GPR43/109a to reduce RAAS activation and cardiac *Nppb* expression, a marker of cardiac hypertrophy and heart failure [30,38]. Alternatively, SCFAs act directly on the vascular endothelium to reset the vascular tone. GPR41 activation in vascular endothelium enhances vasodilation and nitric oxide synthesis; however, GPR41-KO studies suggest possible NO-independent effects [15,37]. This discrepancy may reflect species- or tissue-specific responses. Additionally, SCFAs can modulate gut–CNS signaling. Butyrate or GPR41/Olfr78 silencing reduces inflammation and sympathetic outflow in the hypothalamic paraventricular nucleus, while colonic acetate infusion in sleep apnea-induced HTN lowers neuroinflammation, microglia activation, and BP [32,34].

Several limitations warrant discussion. First, microbial shifts depend on the DF’s physicochemical properties and the baseline microbiome. This review focused on prebiotics, defined as selectively fermented substrates that confer health benefits [13]. Thus, the benefits cited here may not extend to all DFs. Additionally, moderate residual heterogeneity (*I*^2^ = 46%) persisted after controlling for population type, suggesting variable individual responses. In the secondary analysis of adults at high CVD risk, only 40% responded to prebiotics, with responders showing higher baseline levels of the family Lachnospiraceae and the genera *Faecalibacterium* and *Blautia* [46]. Similarly, in an RCT of untreated HTN patients, prebiotic response was predicted by higher baseline levels of the genera *Coprococcus*, *Bifidobacterium*, and *Ruminococcus* [47]. These findings suggest that the response to prebiotics is not global, and patients with a higher baseline abundance of SCFA-producing taxa are more likely to benefit from prebiotic intervention. Second, clinical evidence on SCFA exposure remains limited and inconsistent. Only three studies reported SCFA levels in humans. Interindividual variability in microbiota, driven by diet, ethnicity, and host genetics, limits interpretation of compositional data. However, SCFA levels may serve as a more consistent functional biomarker. Plasma SCFAs reflect systemic exposure, while fecal levels reflect colonic production or absorption deficits; both should be reported when feasible. Third, the variability in prebiotic dose and fiber consumption among human studies complicates the identification of “optimal dosing.” Although researchers attempted to minimize dietary differences between groups, some discrepancies remain unaccounted for. A crossover design may help mitigate these differences. Fourth, most clinical trials had small samples and short durations, limiting generalizability. Larger, longer trials with longitudinal microbiome and metabolome profiling are needed to characterize prebiotics effects over time. Fifth, most included studies relied on 16S rRNA sequencing, limiting taxonomic resolution to the genus level. This is a key limitation, as species within the same genus may differ in function and relevance to BP regulation. Finally, although this review focused on the cardiovascular benefits of prebiotic interventions in the context of HTN, it is important to recognize the global effects of prebiotic-induced modulation of the gut microbiome. For example, butyrate plays a key role in reducing the risk of intestinal diseases such as inflammatory bowel disease and colorectal cancer by enhancing epithelial barrier integrity, suppressing inflammation, and inducing apoptosis in malignant colonocytes [48]. Additionally, restoration of butyrate levels has been shown to improve blood glucose regulation through modulation of glucagon-like peptide-1 (GLP-1) and peptide YY (PYY) secretion, supporting broader benefits in metabolic disease [49]. These shared pathways underscore the potential of microbiome-targeted interventions to reduce chronic disease risk across multiple organ systems.

## 5. Conclusions

Prebiotic supplementation is a promising accessible strategy to maintain BP homeostasis. Clinical and preclinical evidence supports its efficacy in modulating gut microbial composition and enhancing SCFA production. SCFAs act along gut–organ axes to regulate immune, vascular, and neurohormonal pathways involved in BP control. These effects are most pronounced in individuals with preexisting HTN and a higher SCFA-producing capacity. However, interindividual variability in microbial profiles and SCFA exposures limit generalizability. Future research should prioritize characterizing temporal microbiome dynamics in HTN and identifying microbial predictors of response. Dose-finding trials and the development of diagnostic biomarkers to quantify SCFA-producing potential may enable stratified, microbiome-guided interventions.

## Figures and Tables

**Figure 1 nutrients-17-02502-f001:**
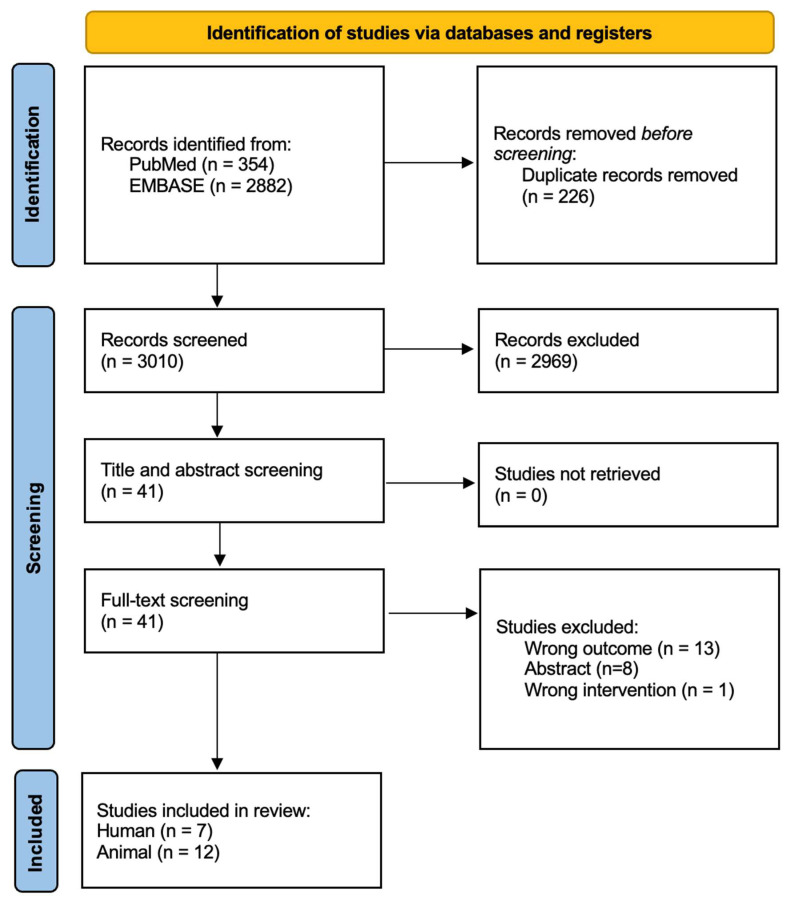
Study selection flow diagram. From PRISMA 2020 statement: an updated guideline for reporting systematic reviews [19].

**Figure 2 nutrients-17-02502-f002:**
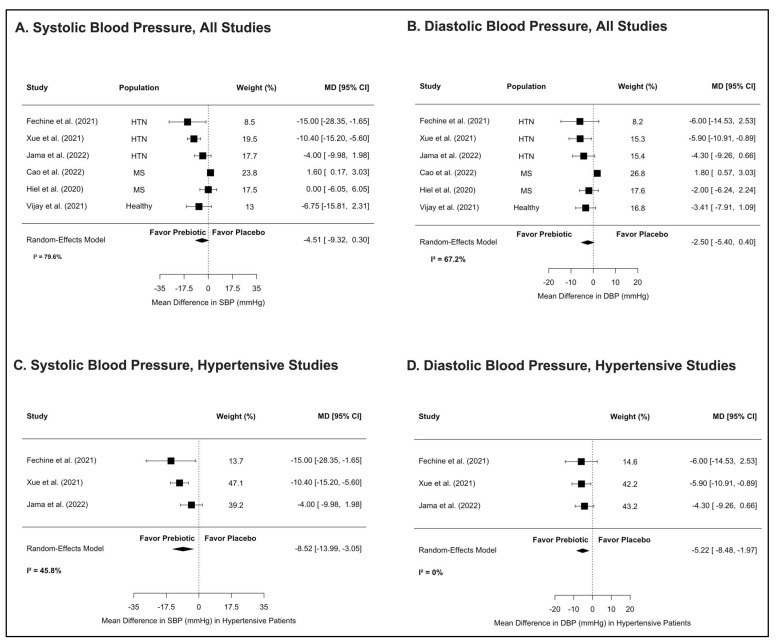
Summary forest plots of mean difference in blood pressure (mmHg) from clinical trials. The upper panel shows the mean differences in systolic (**A**) and diastolic (**B**) blood pressure (BP), along with 95% confidence intervals (CIs), for six clinical studies. The lower panel (**C**,**D**) shows the subgroup analysis restricted to studies including hypertensive patients only (*n* =3) [22,23,24,25,26,28].

**Figure 3 nutrients-17-02502-f003:**
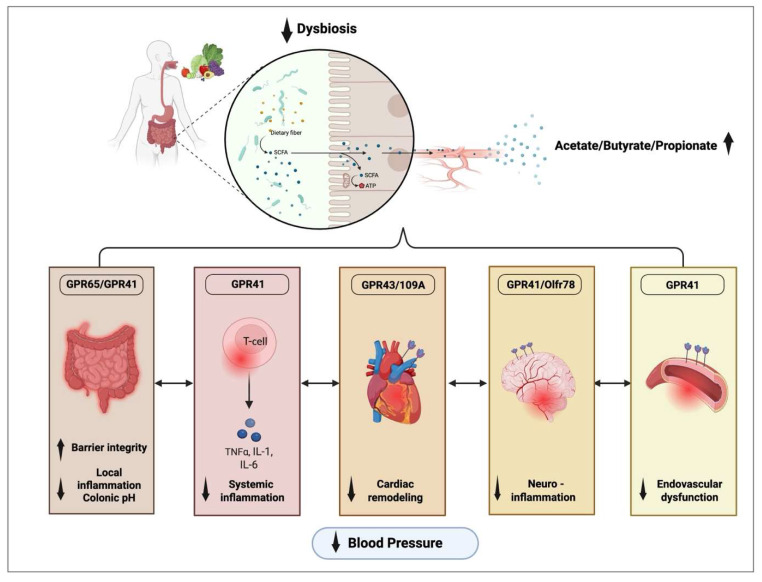
Fiber modulates hypertension pathologies through the gut–organ axis. Indigestible dietary fiber is fermented into short-chain fatty acids (acetate, butyrate, propionate), which regulate BP via gut–organ axes through GPRs (G-protein coupled receptors).

**Table 1 nutrients-17-02502-t001:** Summary of human studies.

Design, Year (*n*)	Population	Intervention	Dose (g/day)/ Duration (Weeks)	Effect on BP	Increase in Main Bacterial Taxa	Increase in SCFA (Site)	Effect on Inflammation/Gut Integrity	Effect on Cardiac/Metabolic Function
RCT, 2021 (14). [22]	20–50 YO **hypertensive** women with obesity (BMI ~ 35)	Mix of soluble/ Insoluble fibers	12/8	Decreased	NR	Butyrate (plasma)	NR	Increased HDL-c, choline, and hydroxybutyrate
RCT, 2017 (44). [23]	18–65 YO **hypertensive** patients (BMI ~ 25)	Oat Bran ^a^	8.9/12	Decreased	*Bifidobacterium_g*	NR	NR	NR
RCT, cross-over, 2022 (20). [24]	18–70 YO untreated **hypertensive** patients (BMI ~ 26)	RS	40/3	Decreased	*Parabacteroides distasonis* & *Ruminococcus Gauverauii*	Butyrate (plasma)	No change/NR	Decreased peripheral resistance
RCT, cross-over, 2022 (27). [25]	18–50 YO with metabolic syndrome (BMI ~ 35)	RS ^a^	20/2	Decreased ^b^	Lachnospiraceae_f & Ruminococcaceae_f	Acetate (fecal)	NR/Decreased endotoxins	Improved glucose/insulin profile ^b^
RCT, multicenter, 2020 (106). [26]	18–55 YO with obesity (50% with diabetes) (BMI ~ 37)	Inulin ^a^	16/12	Decreased ^b^	Actinobacteria_p, *Bifidobacterium_g* & *Anaerostipes hadrus*.	NR	No change/NR	Decreased weight and insulin ^b^
RCT, 2017 (59). [27]	33–55 YO overweight with diabetes (BMI ~ 31)	Inulin	10/6.4VLD	Decreased	*Akkermansia muciniphila*	NR	Decreased HS-CRP and TNF-alpha/NR	NR
Secondary analysis, 2021(69). [28]	>18 YO healthy volunteers with low fiber consumption (BMI ~ 27)	Inulin	20/6	Decreased ^c^	Lachnospiraceae_f, Ruminococcaceae_f, *^d^ Bifidobacterium_g*, *and Coprococcus 3*	^d^ Butyrate (plasma)	Decreased IL4 and TNF-alpha/NR	Decreased GlycA, LDL, VLDL-c and cholesterol

Studies in bold evaluated BP as the primary outcome. ^a^ Both arms received dietary advice. ^b^ In the intervention and control arms. ^c^ Effect lost when adjusting for age, gender and BMI. ^d^ Negatively associated with inflammatory and atherogenic markers. YO: year old; NR: not reported; RCT: randomized controlled trial; BMI: body mass index; RS: resistant starch; BP: blood pressure; SCFA: short-chain fatty acids; HDL: high-density lipoprotein. High-sensitivity CRP: C-reactive protein; GlycA: a novel marker of inflammation associated with cardiovascular incidents; VLDL-c: very low-density lipoprotein. Taxonomic labels are abbreviated as follows: -p (phylum), -c (class), -f (family), -g (genus); species names are written in full.

**Table 2 nutrients-17-02502-t002:** Summary of animal studies.

Model, Year	Intervention Type	Intervention (Duration in Weeks)	Effect on BP	Shifted Bacterial Composition (ß-Diversity) ^a^	Effect on F/B Ratio	Increase in Main Bacterial Taxa	Decrease in Main Bacterial Taxa	Increase in SCFA (Site)	Axis
Angiotensin-II, 2024. [29]	Prebiotic	RS (4)	Decreased	No	NR	*Bacteroides caecimuris*	NA	NR	NA
Angiotensin-II, 2020. [30]	Prebiotic	RS (7)	Decreased	Yes	Decreased	*MacelliBacteroides_g*	*Clostridium* spp.	NR	Gut–Heart
Angiotensin-II + Gpr65 KO, 2022. [31]	Prebiotic	RS (3)	Decreased ^b^	NR	NR	Lachnospiraceae_f and *Bacteroides_g*	*Alistipes-g*	Acetate/propionate (cecal)	Gut–Heart and Gut–Kidney
HFD-programmed fetal HTN, 2022. [32]	Prebiotic (given to offspring)	ITF (6)	Decreased	NR	NR	NR	NA	Butyrate (plasma/PVN)	Gut–CNS
HFD-programmed fetal HTN, 2018. [33]	Prebiotic (given to mother)	ITF (pregnancy/lactation)	Decreased	NR	No change	*Akkermansia muciniphila*	*Bacteroides acidifaciens & Prevotella albensis*	Propionate (plasma)	Gut–Kidney
Hypoxia + high fat diet, 2018. [34]	Prebiotic	RS (2)	Decreased	Yes	NR	Actinobacteria_p, *Bifidobacteria_g*, *Ruminococcus_g*, *& Blautia_g*	Verrucomicrobia_P *& Akkermansia muciniphila*	Acetate (cecal/portal blood)	Gut–CNS
Mild, intermittent hypoxia ^c^, 2020. [35]	Prebiotic	ITF (4)	No change	Yes	NR	*Bifidobacterium animalis*	NA	Acetate (fecal)	NA
Metabolic HTN + ischemic injury ^d^, 2024. [36]	Symbiotic	ITF + probiotic ^e^ (8)	Decreased ^f^	NR	NR	Actinobacteria_p *&* Bacteroidetes_p	Proteobacteria_p	NR	Gut–Heart
Metabolic HTN, 2022. [37]	Prebiotic	DOPS (7)	Decreased	NR	Decreased	Lachnospiraceae_f *& Lactobacillus_g*	*Blautia_g*	Acetate/butyrate/propionate (colon/aorta)	Gut–Heart
Salt-sensitive HTN ^g^, 2017. [38]	Prebiotic	RS (6)	Decreased	Yes	Decreased	*Bacteroides acidifaciens & Bifidobacterium_g*	Enterobacterales_o *& Prevotella_g*	NR	Gut–Heart
Salt-sensitive HTN, 2019. [39]	Prebiotic	Lactulose (4)	Decreased	NR	NR	*Bifidobacterium_g & Subdoligranulum_g*	*Alistipes_g*	NR	Gut–RAAS
SHR, 2021. [40]	Prebiotic	PAO (6)	Decreased	Yes	Decreased	Ruminococcaceae_f & *Bacteroides uniformis*	*Prevotella 9*	No change	Gut–RAAS and Gut–Heart

^a^ Measured using unweighted/weighted UniFrac indexes. ^b^ Gpr65-dependent effect. ^c^ Was not sufficient to disrupt baseline gut microbial composition or function. ^d^ Ischemic injury induced by hyperglycemia and high-fat diet. ^e^
*Lactobacillus* spp./streptococcus/*Bifidobacterium* spp. ^f^ Only in the standard diet arm. ^g^ DOCA model simulated high salt/water phenotype. BP: blood pressure; F/B: Firmicute/Bacteroidetes ratio; SCFA: short-chain fatty acids; NR: not reported; NA: not applicable; RS: resistant starch; KO: knockout; HFD: high-fructose diet; HTN: hypertension; ITF: inulin-type fructans; PVN: paraventricular nucleus; DOPS: dendrobium officinale polysaccharide; RAAS: renin angiotensin-aldosterone system; PAO: potassium alginate oligosaccharides; SHR: spontaneous hypertensive rat model. Taxonomic labels are abbreviated as follows: -p (phylum), -c (class), -o (order), -f (family), -g (genus); species names are written in full.

## Data Availability

Additional data are available on request from the corresponding author.

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
