# Peer review of "Prebiotics Improve Blood Pressure Control by Modulating Gut Microbiome Composition and Function: A Systematic Review and Meta-Analysis"

_nutrients, 2025, doi:10.3390/nu17152502_

Round 1
Reviewer 1 Report
Comments and Suggestions for Authors
This meta-analysis provides the largest scale views from human and animal studies between prebiotics intervention and the control of high blood pressure mediated by gut microbial alterations, mostly focused on SCFA role. The conclusion supported the usage of prebiotics would be beneficial to the control or intervention of abnormal blood pressures. However, several improvements to this draft are still needed to reach the requirement of publication.
Major comments:
- This meta-analysis only repeats the conclusion of single studies but does not improve known cognition.
- Why only focus on the role of SCFAs? The link between prebiotics usage and the intervention of blood pressure should be involved in various metabolites of gut-bacteria derived. This analysis cannot neglect this problem.
- Lipid and bile acid metabolisms are closely related to the disorder of blood pressure. Please add the related advancement, although the studies selected as this meta-analysis may not perform non-SCFA investigations.
- The species or strain-level microbial taxa linked to the control of blood pressure should be shown because not all species or strains of those taxa such as the genera Coprococcus, Bifidobacterium, Ruminococcus, Faecalibacterium, and Blautia can produce potential prebiotics.
- Mechanical studies should be added to this meta-analysis to strengthen the common or novel conclusions.
Need improvement
Author Response
Thank you for your important comments. Please see the attachment.

Reviewer 2 Report
Comments and Suggestions for Authors
The methodology section lacks sufficient detail regarding the precise inclusion/exclusion criteria, and the justification for study selection. The authors mention identifying nearly 3.010 records, yet only 41 studies were ultimately included. This significant narrowing requires clear explanation and transparency—for example, a detailed PRISMA flow diagram, reasons for full-text exclusions, and elaboration on selection filters (e.g., publication type, study design, quality assessment). Without this, the risk of selection bias remains high, and the reproducibility of the review is compromised.
Figure 1, which illustrates the study flow chart and methodological design, should be enriched with more detail (e.g., screening phases, inclusion/exclusion steps, reasons for exclusions). Additionally, it should be relocated to appear immediately after the “Materials and Methods” section and before the Results, as this is the logical and standard placement for study design figures in research articles.
Important variables such as BMI, physical activity, smoking status, family history of colorectal cancer, and comorbidities are not sufficiently described or accounted for in the analysis. A detailed table of baseline characteristics should be included, and the authors should clearly explain how these variables were handled statistically (e.g., adjusted for in regression models
The categorization of dietary intake (e.g., tertiles, quartiles) is inconsistent and insufficiently explained. Additionally, some results—such as the lack of association with low vegetable intake—are reported without adequate biological rationale or discussion. The authors should ensure consistent and clearly defined categorization of all dietary variables, and provide literature-supported interpretation, especially for non-significant or unexpected findings.
Although this is an observational study, the interpretation of dietary associations with colorectal cancer would be significantly strengthened by discussing underlying biological mechanisms. Currently, the manuscript lacks reference to relevant biomarkers or pathways (e.g., chronic inflammation, insulin resistance, oxidative stress) that could help explain how specific dietary components may influence CRC risk. The authors must expand the discussion to include mechanistic insights from the literature, which would enhance the scientific depth and contextualize their findings within the broader field of nutritional epidemiology.
Table 1 and text inconsistently refer to bacterial taxa, mixing taxonomic levels such as phyla, families, genera, and species without proper distinction. This approach is incorrect and may confuse readers. The authors should clearly specify the taxonomic level being reported in each case and ensure consistency throughout the manuscript, particularly in tables and figure legends. Proper labeling (e.g., Genus Streptococcus, Phylum Firmicutes) will improve scientific accuracy and clarity.
As expected for a submission to a peer-reviewed journal, all figures should be clear, high-resolution, and fully legible. However, Figure 2 lacks sufficient clarity—text is blurred or unreadable, making interpretation difficult. The authors should ensure that all visual elements are submitted in publication-quality resolution, with clear legends, labels, and proper formatting to meet the standards of scientific publishing.
Author Response

(The authors gave the same response as above.)

Round 2
Reviewer 1 Report
Comments and Suggestions for Authors
Many thanks for your efforts. My all concerns have been answered.
Author Response
Thank you!
Reviewer 2 Report
Comments and Suggestions for Authors
Thank you for your response. However, I would like to clarify and reinforce the rationale behind my original comment, regarding colorectal cancer.
Although your systematic review and meta-analysis focuses on the effects of prebiotic interventions on blood pressure (diastolic and systolic), the biological role of the gut microbiota and its modulation by prebiotics extends well beyond cardiovascular parameters. Specifically, the gut microbial composition, SCFA production, and epithelial barrier integrity—which are central elements in your mechanistic framework—are also critically involved in colorectal carcinogenesis.
There is substantial literature indicating that microbial dysbiosis, low SCFA levels (especially butyrate), chronic inflammation, and impaired mucosal immunity are shared mechanisms that link altered gut flora not only with hypertension but also with colorectal cancer (CRC), including rectal cancer. Indeed, several of the bacterial taxa discussed in your paper (e.g., Bifidobacterium, Akkermansia, Faecalibacterium, Coprococcus) have dual roles in both blood pressure regulation and colon tumor suppression.
Therefore, including a brief contextual reference to CRC-relevant literature would not fall outside the scope of your review. Rather, it would strengthen your mechanistic discussion by:
(a) Illustrating the pleiotropic effects of prebiotic-induced microbial modulation; (b) Situating your findings within a broader framework of preventive health and chronic disease risk reduction.
Thus, I respectfully maintain that referencing CRC-related mechanisms would improve the discussion, even if colorectal cancer was not directly studied.
Author Response
Thank you for this thoughtful clarification. We agree that highlighting the global effects of gut microbiota modulation enhances the relevance and translational value of our findings. Accordingly, we have added a brief statement to the discussion section referencing the overlapping mechanistic pathways between hypertension and other chronic disease, including CRC, particularly those involving SCFAs, inflammation, and epithelial integrity.
Revised text added to the discussion:
“Finally, although this review focused on the cardiovascular benefits of prebiotic interventions in the context of HTN, it is important to recognize the global effects of prebiotic-induced modulation of the gut microbiome. For example, butyrate plays a key role in reducing the risk of intestinal diseases such as inflammatory bowel disease and colorectal cancer by enhancing epithelial barrier integrity, suppressing inflammation, and inducing apoptosis in malignant colonocytes.1 Additionally, restoration of butyrate levels has been shown to improve blood glucose regulation through modulation of glucagon-like peptide-1 (GLP-1) and peptide YY (PYY) secretion, supporting broader benefits in metabolic disease. 2 These shared pathways underscore the potential of microbiome-targeted interventions to reduce chronic disease risk across multiple organ systems.”
Response also attached
